**TOPICAL REVIEW**

# Post-stroke fatigue – a multidimensional problem or a cluster of disorders? A case for phenotyping post-stroke fatigue

Annapoorna Kuppuswamy [ORCID]

*School of Biomedical Sciences, University of Leeds, Leeds, UK*

Handling Editors: Laura Bennet & Richard Carson

The peer review history is available in the Supporting Information section of this article (https://doi.org/10.1113/JP285900#support-information-section).

**Abstract figure legend** This diagram summarises the many dimensions of fatigue, its behavioural presentation, the multiple models that have attempted to explain fatigue, the winning model of sensory attenuation model, which provides a basis for identifying phenotypes of fatigue along with the need for tailoring treatment to the presenting symptom.

**Annapoorna Kuppuswamy** is a clinical neuroscientist with training in physiotherapy and sensorimotor neurophysiology based at the School of Biomedical Sciences, University of Leeds. Her work focuses on understanding the neural mechanisms of chronic fatigue, and she proposed the Sensory Attenuation Model of fatigue. Her lab's experimental work in post-stroke fatigue bridges lived experience, behavioural manifestations and neural dysfunction. Her lab is also developing brain stimulation-based, phenotype-specific interventions for post-stroke fatigue.

**Abstract** Post-stroke fatigue is a chronic problem with significant impact on morbidity and mortality, which urgently needs effective treatments. The last decade has seen a considerable increase in interest in understanding the pathophysiology of fatigue and developing treatments. In this review, following a summary of theoretical frameworks to understand chronic fatigue, I make a case for why phenotyping fatigue is a necessary step to fully understand pathophysiology, which in turn is essential for the development of robust treatments. I then appraise current post-stroke fatigue literature with the view of identifying post-stroke fatigue phenotypes.

(Received 26 February 2024; accepted after revision 10 October 2024; first published online 30 October 2024)

**Corresponding author** A. Kuppuswamy: Faculty of Biological Sciences, University of Leeds, Room 6.108a, Astbury building, Leeds, LS2 9JT, UK. Email: A.Kuppuswamy@leeds.ac.uk

## – Definitions

Definitions of fatigue: Fatigue is a multidimensional motor-perceptive, emotional and cognitive experience (Annoni et al., 2008). Fatigue is a feeling of lack of energy, weariness and aversion to effort (Mead et al., 2007). Fatigue is a decrease or loss of abilities associated with a heightened sensation of physical or mental strain, even without conspicuous effort, an overwhelming feeling of exhaustion, which leads to inability or difficulty to sustain even routine activities and which is commonly expressed verbally as a loss of drive (Staub & Bogousslavsky, 2001).

Operational definition of fatigue: A percept arising primarily from alterations within the activational systems that inform voluntary action

Motor trigger: A neuromuscular state that is likely to result in fatigue.

Cognitive trigger: A cognitive state likely to result in fatigue.

Physical fatigue: A loosely defined term largely associated with motor triggers, and describes a reduction in motor output, normally brought on by repeated muscular contractions, sometimes used interchangeably with 'fatigability'.

Mental fatigue: A loosely defined term related to cognitive triggers and describes a feeling of inability to continue performing an activity predominantly thought to be a result of diminished motivation, sometimes used interchangeably with 'fatigue'.

Fatigability: The property of a system to diminish its output with repeated activation of the system.

Acute fatigue: Where fatigue is triggered by a well-defined stimulus such as a stroke, infection or prolonged activity, fatigue is experienced within the time window closest to the initial trigger. The most used acute time window is up to 3 months from the trigger, with 3–6 months regarded as sub-acute. For activity-induced fatigue, the acute time window is much shorter, in the order of hours. Acute fatigue is reversible.

Chronic fatigue: Fatigue present in chronic illnesses and is thought to be associated with established disease processes. A clear time frame with regard to the initial trigger is lacking and is still a matter of debate. For the purposes of investigation, anything that falls outside the acute time window has been used to identify chronic fatigue. Chronic fatigue is largely irreversible.

Primary fatigue: Fatigue attributable directly to the primary disease pathology.

Secondary fatigue: Fatigue attributable to comorbid conditions, or not directly linked to the primary disease pathology. For example, in a stroke survivor with comorbid anaemia, fatigue arising from the stroke is primary, while anaemia-related fatigue is secondary. While primary and secondary fatigue are indistinguishable from a phenomenology perspective, they can be distinguished by treating the comorbid condition to identify the source of fatigue.

## Introduction

Fatigue is a common symptom of uncertain origins in many chronic diseases (Chaudhuri & Behan, 2004; Davies et al., 2021; Sleight et al., 2022), including in those living with the consequences of stroke. The lack of a clear definition, and the dynamic multidimensional nature of the symptom has hindered progress in understanding the physiology of fatigue. The multidimensional aspect of fatigue is more strongly manifested in conditions such as stroke. Despite a common aetiopathology, the location of the stroke determines post-stroke deficits, giving rise to a wide range of post-stroke sequelae, including multiple types/multidimensional fatigue. Most definitions of fatigue refer to a perceptual experience. However, the term has also been used to describe changes in

neuromuscular functioning following repeated muscular contractions, i.e. fatigability (see Box 1 for definitions of fatigue and the various terminology associated with fatigue). In this paper, the term 'fatigue' refers to a perceptual experience (see Box 1 for the operational definition of fatigue). Most experimental paradigms rely on repeated muscular contractions to investigate both fatigability and fatigue (Enoka & Duchateau, 2016), with far fewer studies investigating non-exercise induced fatigue, which is the predominant type of fatigue in pathological states. Attempts to highlight the significance of precise use of terminology (Kluger et al., 2013; Skau et al., 2021) and the means to distinguish fatigue from fatigability (Tankisi et al., 2024), help refocus our efforts towards understanding this perceptual phenomenon.

Multidimensionality is a well-acknowledged feature of fatigue, a feature attributable to multiple experiences reported by those with fatigue (Whitehead et al., 2016). Over the last few decades, robust questionnaires have been developed to capture all aspects of the multidimensional nature of fatigue (Dittner et al., 2004; Hewlett et al., 2011). However, the origin of this multidimensional symptom is far from clear. In diseases such as anaemia where the primary pathology (metabolic disturbance) can fully explain fatigue (single-source origin), the origin is less ambiguous than in conditions such as stroke, and other diffuse central and peripheral nervous system disorders, where the multidimensional nature of fatigue is more strongly manifested. Despite the presence of a range of biological (Akcali et al., 2017; Booij et al., 2018; De Doncker, Brown, & Kuppuswamy, 2021; Kuppuswamy et al., 2015; Kutlubaev et al., 2012; Morris et al., 2015; Ondobaka et al., 2021; Su et al., 2014) and psychosocial (Cumming et al., 2018; Naess et al., 2012) correlates of fatigue in stroke and other neurological disorders, and several theoretical models of fatigue (Dantzer et al., 2014; Dobryakova et al., 2015; Kuppuswamy, 2017; Sleight et al., 2022; Stephan et al., 2016; Tanaka et al., 2013), there is still a need for more complex models to explain how multiple experiences emerge from biological and psychosocial factors. Perhaps it is time to reduce the dimensionality of the problem by categorising fatigue into phenotypes. Phenotypes may reveal non-overlapping characteristics, whose pathophysiology is easier to understand without the need for increasingly complex models. This is particularly important in neurological disorders where the number of investigations targeting neural network dysfunction is on the rise. Without classification into phenotypes, we run the risk of identifying too many fatigue-related neural network dysfunctions which most likely underpin different fatigue-related experiences, and are not equally generalisable across all individuals with fatigue. Wide-ranging associations between fatigue and neural network dysfunction render cross-comparisons between studies difficult, which, in the worst-case scenario is attributed to poor methodology or, in the best case, attributed to differences in patient selection, fatigue scale used, etc. Phenotyping fatigue will provide a systematic method for identifying homogeneous patient groups to increase the chances of understanding the underlying pathophysiology of the multiple presentations of fatigue. Phenotyping is particularly pertinent in conditions such as stroke which, from the perspective of stroke sequelae, is a group of disorders rather than a single condition. A natural progression of this idea is that treatments for fatigue and post-stroke fatigue must also be tailored to the presenting symptoms, with the tailoring rendered significantly easier with a knowledge of fatigue phenotypes.

**Dimensions of fatigue.** Physical and mental fatigue, broadly referring to 'fatigability' and 'fatigue', respectively, are the most identifiable dimensions entrenched within both common parlance and scientific literature. However, I have discussed elsewhere how this terminology is problematic and that physical and mental dimensions must be understood as motor and cognitive triggers for fatigue (Tankisi et al., 2024) (see Box 1 for definitions). Questionnaires that capture dimensions of fatigue (Learmonth et al., 2013; Smets et al., 1995) support the notion of motor and cognitive triggers as being two separate dimensions in addition to other dimensions such as motivational and psychosocial dimensions. While motor triggers are often understood as arising from repeated muscular contractions, cognitive triggers are understood as arising from a reduction in motivation resulting from repeated deployment of a resource (motor or cognitive). However, such an understanding is largely driven by the vast exercise physiology literature (Enoka & Duchateau, 2016; Enoka et al., 2011) that focuses only on non-pathological fatigue states where there is a certain degree of interconnectedness between motor and cognitive triggers, due to a common denominator, i.e. repeated activation questionnaires that measure pathological fatigue make a distinction between motor, cognition and motivational dimensions. This distinction assumes that they can vary independently. This assumption arises from a need to account for motor, cognitive and motivational deficits which in themselves contribute to fatigue without repeated activation resulting in changes in all three systems. In the absence of any clear theoretical frameworks for explaining how, or which, motor and cognitive deficits might result in fatigue, pathological fatigue has, until recently, largely relied on inflammation-induced motivational blunting (sickness behaviour) as its sole explanation (Dantzer et al., 2014). However, this only explains the motivational dimension of fatigue and fails to explain the motor and cognitive dimensions.

**Theoretical basis of fatigue.** Fatigue has been acknowledged as a phenomenon worthy of scientific enquiry for over a century. However, fatigue in disease conditions (pathological fatigue), until recently, received little attention. Here, I briefly summarise eight theoretical models of pathological fatigue and compare them for their ability to explain multiple dimensions of fatigue.

Neuroimmune model of fatigue (Dantzer et al., 2014): The fundamental premise of this model is that inflammatory mediators released by activated innate immune cells at the periphery and in the CNS profoundly disturb the neuronal environment. The resulting alterations in fronto-striatal networks together with the activation of the insula explain the many dimensions of fatigue, including reduced incentive motivation, decreased behavioural flexibility, uncertainty about the usefulness of actions and awareness of fatigue.

Work output model of pathological fatigue (Chaudhuri & Behan, 2004): Pathological fatigue is best understood as an amplified sense of normal (physiological) fatigue that can be induced by changes in one or more variables regulating work output. Work output is a dependent variable of applied effort that is controlled by motivational input (internal and external) and feedback from motor, sensory and cognitive systems that establish the level of perceived exertion.

Sensory attenuation model of fatigue (Kuppuswamy, 2017, 2022): Fatigue is an inference of high perceived effort, underpinned by poor attenuation of anticipated sensory inputs from peripheral motor apparatus (proprioceptive sensory inputs). A second iteration of this model explains how poor attenuation of other types of sensory input, such as exteroceptive and interoceptive inputs, also results in high perceived effort. The sensory attenuation model uses a single type of deficit, i.e. diminished sensory attenuation (which underpins the perceptual experience of high effort), applied to multiple sensory streams to explain the multiple dimensions of fatigue such as exteroceptive sensitivity, heightened interoceptive awareness and muscular system-related symptoms such as body heaviness (proprioceptive awareness).

The facilitatory–inhibitory conditioning model of fatigue: Fatigue is explained by hyperactivated excitatory and inhibitory systems (Tanaka & Watanabe, 2012) in the brain. This model was developed to explain exercise-induced reduction in corticospinal output to peripheral motor apparatus. Inhibitory factors in the form of sensory input from the periphery and excitatory inputs in the form of motivational drive, when hyperactivated, help maintain performance in the face of fatigue, which then eventually fails, diminishing corticospinal output. In pathological states, chronic fatigue is experienced due to a conditioned hyperactive inhibitory response to minimal exertion. This theory explains activity intolerance in pathological fatigue.

Dopamine hypothesis of fatigue (Dobryakova et al., 2015): Fatigue arises due to an imbalance of dopamine, a modulatory neurotransmitter, in the CNS and the immune system.

Metacognitive theory of dyshomeostasis (Stephan et al., 2016): This is a combined model of fatigue and depression set within a predictive processing framework, focusing on metacognitive processes. In this framework, fatigue and depression can be understood as sequential responses to the interoceptive experience of dyshomeostasis and the ensuing metacognitive diagnosis of low allostatic self-efficacy (Stephan et al., 2016).

Biopsychosocial model (Wade & Halligan, 2017) of fatigue: All the above models are biological models. However, perceptual experiences are context-dependent, with context arising from both within and without the biological system. The biopsychosocial model is an interdisciplinary model that counteracts biological reductionism which relies on the interaction between external and internal context to explain fatigue.

3P model (Wright et al., 2019) of fatigue The 3P model provides greater granularity than the biopsychosocial model by segregating contributing factors into predisposing, precipitating and perpetuating factors to explain how complex symptoms such as fatigue arise from different factors.

**Model suitability for phenotype classification.** A phenotype is defined by a distinguishable set of observable characteristics that arise from the interaction between an organism of a particular biological constitution and its environment. Chronic fatigue, particularly in conditions such as stroke, which can vary significantly based on lesion location (multiple biological constitutions), suggests a likelihood of fatigue phenotypes.

The neuroimmune model explicitly refers to how multiple dimensions of fatigue can be explained by a common mechanism. The neuroimmune model relies on alteration in the neurochemical environment as a basis for fatigue. However, conditions where fatigue is chronic, and inflammatory processes which drive neurochemical imbalances, have all but died down, there is less evidence in support of inflammation-driven fatigue. Assuming the altered neuronal environment continues into the chronic phase, could the neuroimmune model distinguish between phenotypes? Multiple dimensions such as poor behavioural flexibility and reduced motivation depend on a common neural substrate such as the prefrontal cortical circuitry, a super-hub with multiple functions, and therefore would struggle to explain how multiple features can manifest independently of one another, as is required for distinguishable phenotypes. Assuming the many dimensions could be explained by inflammation-induced changes in fully, or partially non-overlapping neural

circuits, it is unclear how a widespread change in neuronal environment can differentially affect different brain circuits. Therefore, the neuroimmune model would strongly support multidimensionality over phenotypes.

The metacognitive theory of dyshomeostasis, although it encompasses multiple manifestations of fatigue, treats all dimensions as being linked to one another, with resulting dyshomeostasis being the primary driver. The assumption of a link between multiple dimensions indicates that this theory would predict a lack of distinguishable phenotypes. The facilitatory–inhibitory conditioning theory explains only a particular dimension of fatigue, i.e. activity intolerance, and it is unclear how multiple dimensions could emerge from maladaptive conditioning. The work output model, despite not explicitly stating the possibilities of phenotypes, refers to independent modulators of perceived effort and thus allows for distinguishable phenotypes to emerge. The dopamine hypothesis model is similar to the neuro-immune model and predicts widespread changes in dopamine function with no clear basis for the emergence of independent phenotypes. The biopsychosocial models, which go beyond biological mechanisms, would subsume the above biological models, and could potentially be used to identify fatigue phenotypes.

The sensory attenuation model explicitly states its ability to explain the diverse phenomenology of fatigue such as that arising from bodily sensations, external stimuli and cognitive difficulties. This model relies on neural network dysfunction with no requirement for neurochemical imbalances. The model proposes altered attentional mechanisms, which could only emerge at the neural network level and do not necessarily require neurochemical imbalance. Attentional mechanisms are implemented by synchronous neural firing across multiple brain regions. The functional nature of the proposed mechanism allows different circuitries to be independently dysfunctional and strongly predicts the presence of phenotypes. The model further identifies three possible phenotypes – the exteroceptive, inter-oceptive and proprioceptive phenotypes which emerge from a common deficit of poor sensory attenuation.

Overall, the sensory attenuation model is the strongest which not only predicts the presence of phenotypes but also predicts what the possible phenotypes are and how phenotypes might emerge.

**Post-stroke fatigue.** Post-stroke fatigue (PSF) is defined as 'a feeling of exhaustion, weariness or lack of energy that can be overwhelming, and which can involve physical, emotional, cognitive and perceptual contributors, which is not relieved by rest and affects a person's daily life' (English et al., 2023). Stroke is a highly heterogeneous condition with the prime driver of heterogeneity being lesion location. Lesion volume or location does not determine the incidence or severity of fatigue (Jolly et al., 2023). However, a greater incidence of fatigue following a stroke, when compared with a transient ischaemic attack, suggests that PSF is a direct consequence of neuro-nal damage and not a non-specific symptom (Winward et al., 2009), especially in the chronic phase. Given that stroke-induced neuronal damage causes PSF, and stroke affects different parts of the brain in different individuals, it logically follows that fatigue arising as a direct consequence of the stroke must have, at least partially, lesion-dependent mechanisms. This would especially be true in the chronic phase post-stroke, when the acute inflammatory neuronal environment is no longer a significant factor for PSF, with neuroplastic mechanisms driving lesion-dependent changes in neural network-level activity. A lack of association between lesion characteristics and PSF indicates there is no 'fatigue centre' in the brain which, when damaged, results in fatigue; instead, a distributed network of regions likely contribute to the development of fatigue.

PSF is seen in stroke survivors immediately post-stroke and for several years after stroke, with documented evidence up to 7 years post-stroke (Christensen et al., 2008; Glader et al., 2002; Pedersen et al., 2022). While acute PSF is not necessarily intrusive, chronic PSF is a major detriment to quality of life, independent living, return to work, risk of developing mental health disorders and results in higher mortality. Therefore, it is more important to tackle chronic PSF than acute PSF. While it may be argued that acute PSF could lead to chronic PSF, and tackling acute PSF is more important, there is evidence to suggest the existence of late-onset PSF with no acute PSF (Wu, Mead, et al., 2015). Moreover, the contribution of different dimensions of fatigue to the over-all PSF level changes from acute to chronic phase (M. Delva et al., 2018), indicating an intriguing possibility that in those who present with both acute and chronic PSF, the mechanisms of acute and chronic PSF may be different. In the acute phase, there is evidence of links between inflammatory environment and PSF severity which all but disappears around 6 months post-stroke (Ormstad et al., 2011; Wu, et al., 2015).

**Proposed PSF phenotypes.** In the chronic phase, a range of behavioural and neural network dysfunctions characterise PSF. The sensory attenuation and 3P models provide the most scope for explaining the many experiences and behavioural presentations associated with PSF, making them the most suited for identifying phenotypes of PSF. A recent proposal of a 3P model for PSF, known as the unifying model of PSF (Kuppuswamy et al., 2024), elaborates how the neuroimmune (early first-order precipitating factor), sensory attenuation (late

first-order precipitating factor) and metacognitive models (second-order precipitating) can all contribute to PSF at different time points post-stroke.

The sensory attenuation principle embedded within the precipitating factors of the unifying 3P model posits that attentional dysregulation within different sensory streams results in three PSF phenotypes. The proprioceptive phenotype arises from a lack of attenuation of self-generated sensory input from muscular contractions presenting as effortful muscle contractions and body heaviness at rest; the exteroceptive phenotype arises from a lack of attenuation of task-irrelevant audiovisual sensory stimuli presenting as audiovisual hypersensitivity; and the interoceptive phenotype arises from lack of attenuation of visceral sensory input manifesting as heightened bodily sensations such as breathlessness, palpitations, etc.; and all of the above fundamental dysfunctions result in a reduced willingness to engage in activity (Fig. 1).

A number of other higher-order cognitive correlates of PSF such as diminished processing speed, reduced executive functioning and poor behavioural flexibility can all potentially be explained by the above fundamental sensory processing attentional deficits. This model fits well with the notion of a distributed network of brain regions, any of which, when affected, results in fatigue, and is

particularly well suited to explain PSF which can arise from a variety of different lesion locations.

**Sensory attenuation, thalamocortical circuits and PSF.** A recent large meta-analysis of associations between lesion location and PSF shows thalamic involvement is necessary for fatigue to manifest in the chronic phase post-stroke, but not in the acute phase (Zhang et al., 2020). Interestingly, the lesioned regions that extend beyond the thalamus do not overlap (Wang et al., 2022), which suggests that while thalamic involvement is necessary for fatigue to manifest, the three phenotypes of PSF are driven by lesions beyond the thalamus itself.

The thalamus is a critical sensory hub with higher-order thalamic nuclei receiving and sending inputs to layer 5 pyramidal neurons across cortical regions, forming discrete thalamocortical loops. Afferent activity in this loop is thought to be central to conscious perception of sensory input and efferent activity provides the context for sensory perception making the thalamocortical loops a suitable substrate for the integration of top-down and bottom-up signals for conscious perception (Takahashi et al., 2020). Any disruption in activity in these loops will result in abnormal sensory experiences. Several altered sensory states, including disorders of consciousness

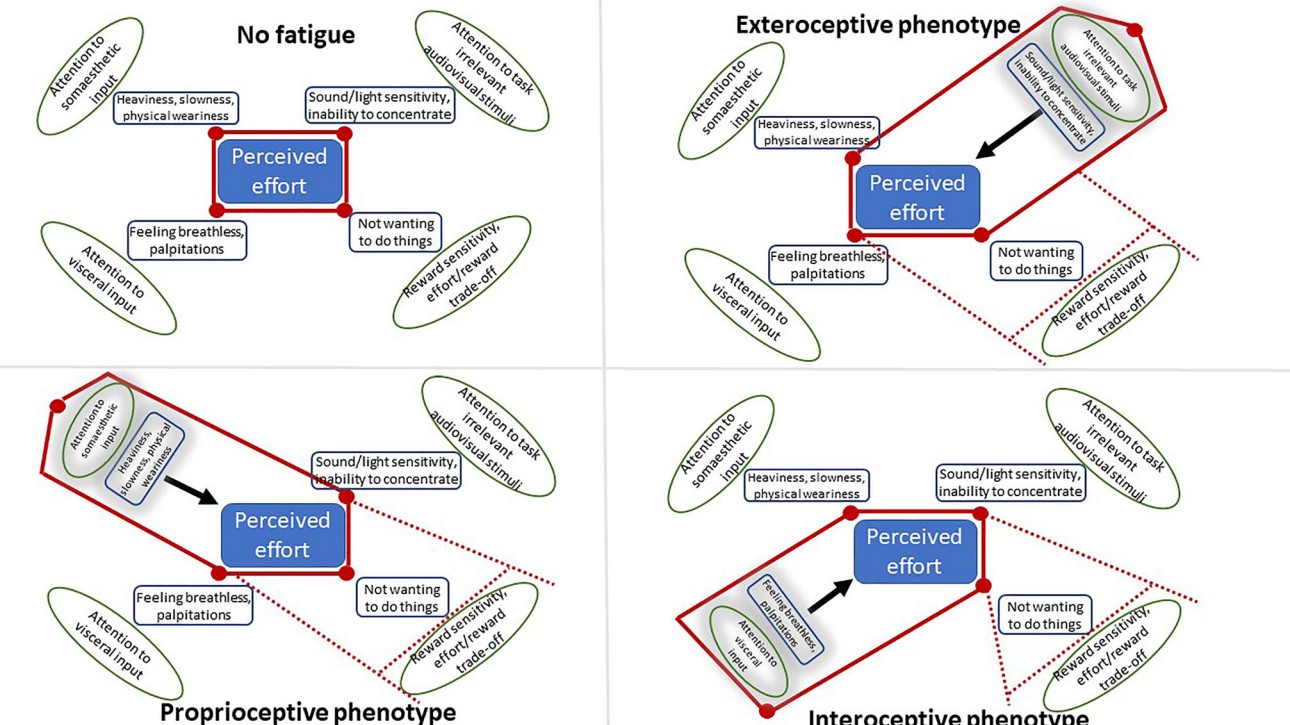

**Figure 1. Sensory attenuation model and phenotypes of post-stroke fatigue**
This schematic shows how attentional dysregulation to different sensory stimuli (green ovals) underpins different perceptual manifestations (blue boxes) that contribute to perceived effort, which is inferred as fatigue. The solid red lines indicate phenotype-specific presentations and underlying dysfunction, while dashed red lines indicate the resultant higher-order dysfunction which is common to all phenotypes.

(Redinbaugh et al., 2020) and affective symptoms such as fatigue (Arm et al., 2019; Capone et al., 2020), have been associated with thalamocortical loop disturbances. The sensory attenuation theory attributes fatigue to abnormal sensory experience driven by poor attenuation (top-down/bottom-up signal integration) of anticipated (top-down attention) sensory input (bottom-up signals), thereby explaining how thalamocortical loop dysfunction is well-positioned to implement poor sensory attenuation across multiple sensory modalities.

The complex blood supply to the different thalamic nuclei that can be independently occluded presenting as different thalamic syndromes (Bordes et al., 2020; Schmahmann, 2003), offers the possibility of phenotype distinctions at the level of thalamic lesions. However, this is purely speculative at this point. For example, lesions within the ventroposterior complex, combined with lesions in extrathalamic somatosensory and motor pathways would give rise to the proprioceptive phenotype of fatigue. Similarly, lesions of the medial geniculate combined with temporal lobe lesions would give rise to the exteroceptive phenotype of fatigue. While the possibility of each thalamic territory giving rise to a separate fatigue phenotype can be entertained at a theoretical level, the phenomenological and behavioural presentation of fatigue (unpublished findings from our laboratory) is primarily associated with altered somatosensory, audiovisual and viscero-sensory experiences, which strongly suggests three phenotypes as proposed above. Neural findings are also confined to these networks.

Thalamus sub-region specific whole-brain functional connectivity showed independent alterations in somatomotor-thalamic and occipito-thalamic connectivity in PSF, with a discussion of how different dimensions of PSF might relate to different thalamo-cortical loop dysfunction (Wang et al., 2024). The instrument used to measure PSF in this study, unfortunately, did not allow measurement of different dimensions of fatigue, to identify dimension-specific thalamocortical dysfunction. Whether such differential manifestations of fatigue are significant enough to allow categorisation of PSF into distinguishable phenotypes, is not known. If distinguishable, at what levels are they distinguishable – self-report/perceptual, behavioural, neural or at multiple levels? What follows is an appraisal of the perceptual, behavioural, neural and interventional literature in PSF to address the question, 'Can PSF be segregated into phenotypes?'

## Perception and behaviour-based phenotypes of post-stroke fatigue

Questionnaires are the gold standard measure of PSF (Bicknell et al., 2022; Eilertsen et al., 2013; Eriksson et al., 2022; Flinn & Stube, 2010; Kirkevold et al., 2012; Thomas et al., 2019; White et al., 2012; Whitehead, 2009) and variability in the behavioural manifestation of PSF has been investigated by few studies. While general fatigue levels relate to both physical and cognitive impairments, physical subdomain fatigue scores more strongly relate to functional disability, and cognitive impairments specifically related to mental speed, working memory and verbal short-term memory, are strongly associated with mental fatigue (M. Delva et al., 2018; Hubacher et al., 2012; Johansson & Rönnbäck, 2012). While functional disability is linked to the physical fatigue score, motor impairments don't explain fatigue, but altered neuromuscular sensations do (Kuppuswamy et al., 2016). Despite parity in motor output, for a given motor output, the experience of effort is higher in those with higher PSF (De Doncker et al., 2020). This suggests that altered muscle contraction-related sensory experience contributes to functional disability in PSF and not motor impairment *per se*. The proprioceptive phenotype of PSF in accordance with the sensory attenuation model would predict that such altered sensory experience drives high perceived effort. Therefore, the presence of such altered muscular sensations in the absence of motor impairment is evidence in favour of sensory attenuation-driven proprioceptive phenotype.

The association between cognitive impairments and the mental fatigue score is harder to interpret within the sensory attenuation model. However, cognitive tests rely on intact audiovisual perception and any distortions in perceptual processing will likely influence the test results. Cortical representation of auditory and visual stimuli, specifically task-irrelevant stimuli (distractors), are not sufficiently attenuated, resulting in sensory overload in PSF (De Doncker & Kuppuswamy, 2024; Kuppuswamy et al., 2022). Assuming that sensory overload is detrimental to the performance of cognitive tasks, one may predict that the association between multiple cognitive deficits and mental fatigue is driven by diminished exteroceptive stimulus attenuation. The exteroceptive phenotype of PSF in accordance with the sensory attenuation model would predict such diminished exteroceptive attenuation.

Complex deficits that involve both physical and cognitive components such as binocular visual dysfunction relate equally to physical and cognitive subdomain scores (Schow et al., 2017). A breakdown of individual components such as oculomotor dysfunction and visual stimulus integration will be a more useful test for purposes of phenotype assignment. However, such multidomain dysfunction may not be separable and could be the basis of a mixed phenotype.

With the passage of time, despite overall fatigue levels being stable, the contribution of mental subdomain scores increases while physical subdomain scores decrease (M.Y.

Delva et al., 2018), which might suggest that a single individual could switch phenotypes depending on time post-stroke. In those who do not present with early fatigue, late-onset fatigue mostly co-occurs with mood-related disorders and not physical or cognitive impairments (M. Delva et al., 2018). Despite co-occurrence, separate mechanisms, at least partially, are thought to under-pin such mood disorders related to PSF. Interventional studies demonstrate that anti-depressants have no effect on PSF (Choi-Kwon et al., 2007; Karaiskos et al., 2012) while reducing depressive symptoms, indicating possible independent mechanisms driving PSF and depression. Such PSF associated with long-standing mood disorders could possibly belong to the interoceptive phenotype driven by long-standing dyshomeostasis.

Depression, anxiety and pain are strongly associated with fatigue (Appelros, 2006; Naess et al., 2012; Ponchel et al., 2015), with depression-associated PSF more likely to score high on the mental and motivational subscales, while physical fatigue scores are more strongly linked to anxiety (Mutai et al., 2017). While the evidence is far from complete, PSF, despite its complex behavioural pre-sentation that could change over time, the above evidence shows links between the perceptual and behavioural pre-sentation of PSF in a manner that lends support to the three proposed phenotypes of the sensory attenuation model of fatigue.

## Neuronal structure and function-based phenotypes of post-stroke fatigue

The literature on lesion location and PSF is extensive with mixed results and has been comprehensively reviewed elsewhere (Jolly et al., 2023). Despite a large volume of investigations dedicated to lesion location, only one has investigated the relationship between fatigue subdomain scores and lesion location (Hubacher et al., 2012). This study demonstrated that while overall fatigue levels were not related to lesion location, those with cortical lesions scored higher on the cognitive subdomain of fatigue, and those with subcortical lesions presented with higher fatigue in the physical domain. From the viewpoint of phenotype assignment, cortical lesions would belong to the exteroceptive phenotype, while subcortical lesions to the proprioceptive phenotype. This is also in line with anatomical substrate that subserve different functions, with somatomotor circuits having a greater representation within the subcortical circuitry.

Interestingly, this investigation used two fatigue questionnaires (Fatigue Scale for Motor and Cognitive Functions (FSMC) and Modified Fatigue Impact Scale (MFIS)), both with physical and cognitive fatigue sub-domains and showed that the differentiation between subdomain scores based on lesion location was only evident using FSMC and not MFIS. FSMC also identified a further 23% more individuals as having clinically relevant fatigue than MFIS, which could be one factor contributing to the difference in associations while using the two scales. This also highlights the importance of using the right instrument to capture the lived experience of targeted deficits.

The recent guidelines for PSF research (English et al., 2023) highlight the extraordinary breadth and variability of content in scales and emphasise the need to carefully choose the scale that is fit for purpose. Thus, the finding from the single small study of lesion-based differentiation of fatigue manifestation is sufficient to fuel more investigations in lesion-based phenotyping of PSF. In addition to carefully choosing the right instrument for quantifying domain-specific fatigue, the methodology for the classification of lesions must also move away from broad categorisations to more sophisticated fine-grained methods such as voxel-based lesion–symptom mapping, and nuanced lesion categorisations (Bonkhoff et al., 2021) for robust lesion-based phenotyping.

While structural damage following stroke is relatively obvious to identify and measure, functional disruption is significantly more difficult to quantify, primarily due to the wide-ranging plastic changes across the entire brain. Since the first glimpse of neural dysfunction in the form of suppressed cortical excitability in PSF nearly a decade ago (Kuppuswamy et al., 2015), there have been a number of investigations to better understand neural dysfunction in PSF. However, none of the investigations have focused on fatigue subdomain-specific differences. Due to the inherent complexity of investigations, and the need to control for significant confounders such as depression, neurophysiology investigations have been performed in non-depressed, minimally impaired stroke survivors. These findings are not conducive to phenotype assessment. However, resting-state and task-related neural activity in PSF would help to understand whether the pathophysiology of PSF would fulfil the predictions of fatigue models.

PSF severity is related to reduced excitability of the corticospinal tract (CST) arising from the left hemi-sphere, but increased excitability of the right CST (De Doncker & Kuppuswamy, 2024). Interhemispheric inhibitory imbalance explains CST excitability findings irrespective of the side of the stroke lesion (Ondobaka et al., 2021). While interhemispheric inhibition appears to be altered, intracortical (within M1) inhibition does not show any abnormalities (Kuppuswamy et al., 2015) while intracortical facilitation is altered (Kindred et al., 2024). These resting-state abnormalities in motor cortical excitability could be attributed equally to inflammation-induced neurochemical changes or diminished sensory attenuation driven by heightened inhibitory drive from regions upstream of the motor

cortex, such as the somatosensory cortex. Resting-state functional connectivity studies in PSF confirm the greater somatosensory activity with a consequent diminished motor cortex connectivity (Wu et al., 2023). A case against neurochemical theory would be: (a) a lack of altered inflammation-driven connectivity in cortical regions outside of the primary motor networks (Ondobaka et al., 2021) (note, non-hypothesis driven whole-brain functional connectivity studies show altered connectivity in non-motor brain regions (Cotter et al., 2022; Ren et al., 2024; Schaechter et al., 2022)) and (b) the chronicity of studied populations where there is limited evidence of ongoing neuroinflammation linked to PSF (Jolly et al., 2024). However, given the increasing evidence in support of continued CNS inflammation in chronic stroke (Ermine et al., 2021; Smith et al., 2013; Wang et al., 2007), there is scope to further explore links between CNS chronic inflammation, immune response and PSF.

Resting-state abnormalities continue into the pre-movement phase and during movement, with diminished premovement inhibition (De Doncker, Brown, & Kuppuswamy, 2021) and a greater drop in the central drive to CST during maximal muscle contraction (Knorr et al., 2011). In submaximal muscle contractions there is asynchronous drive to spinal motor units, which indicates greater inhibition, to CST. Such changes to CST input are reflected in motor behaviour in the form of greater variability in force output (De Doncker et al., 2020). PSF-related differences during movement preparation and muscle contraction can be explained by both the conditioned inhibitory response theory and the sensory attenuation model of fatigue.

Inhibitory input to CST could also arise from other sensorimotor pathways, such as visual and auditory pathways. Our recent findings of poor attenuation of task-irrelevant auditory and visual stimuli in PSF (De Doncker & Kuppuswamy, 2024; Kuppuswamy et al., 2022) support the presence of such inhibitory inputs. While the conditioned inhibitory response theory does not explicitly state what type of inhibitory inputs underlie reduced motor cortical excitability, the sensory attenuation model does, with the above corroborating evidence.

Overall, results from neurophysiological and neuro-imaging studies lend support to multiple models of fatigue, with the sensory attenuation model providing the strongest explanations to directly link lived experience, behavioural manifestation and neural processing in segregated neural networks. Such segregated PSF-related neural dysfunction could form the basis of the three proposed PSF phenotypes. Direct evidence of distinguishable PSF phenotypes based on neuronal structure and function is still lacking, but that is primarily due to the paucity of hypothesis-driven experiments and offers significant scope for future experiments.

## PSF interventions and phenotypes

The most promising pharmacological intervention for PSF is modafinil (Bivard et al., 2017; Brioschi et al., 2009; Poulsen et al., 2015), a weak dopamine reuptake inhibitor. Despite its effectiveness, not all those with PSF respond to modafinil. The strongest predictor of a positive response is diminished functional connectivity between the dorso-lateral prefrontal cortex and caudate nucleus, with those exhibiting normal (similar to non-PSF group) dorso-lateral prefrontal–caudate connectivity not benefiting from modafinil therapy (Visser et al., 2019). This result indicates that there are at least two phenotypes: one that responds to modafinil and another that doesn't. Given the circuitry that predicts responsiveness to modafinil, one might speculate that the responsive phenotype belongs to exteroceptive or interoceptive phenotypes, while the unresponsive belongs to the proprioceptive phenotype.

Transcranial direct current stimulation (tDCS) is a neuromodulation intervention which shows promising results for the reduction of PSF, with two target sites, the motor cortex and dorsolateral prefrontal cortex, both effective at a group level. The motor site response is predicted by anxiety levels and mediated by a change in the gain function of corticomotor networks in non-depressed, minimally impaired stroke survivors (De Doncker, Ondobaka, & Kuppuswamy, 2021). The motor responsive phenotype is characterised by an absence of mood disorders (depression and anxiety) and any motor or cognitive impairment. Given the mechanism of action of motor tDCS, i.e. reduction of motor cortex input gain, inputs that partly arise from somatosensory cortices, the motor tDCS responsive phenotype exhibits characteristics of the proprioceptive phenotype. There is insufficient information in dorsolateral prefrontal cortex neuromodulation studies (Dong et al., 2021; Ulrichsen et al., 2022) to identify any predictors of response to stimulation, for phenotype assignment.

A recent attempt at phenotyping PSF used the k-means clustering method on a limited number of variables, including measures of stroke characteristics, mood and motor impairment. Despite the limitations of a number of variables and the limited range of each available variable, this study identifies four distinct phenotypes of PSF (De Doncker & Kuppuswamy, 2024). The cluster characteristics are low mood/low function, low mood/high function, high mood/low function, and high mood/high function. The high-functioning clusters were exclusively left-hemispheric strokes, while low-functioning clusters were exclusively right-hemispheric strokes. There was no significant difference between fatigue levels across the four clusters. This study demonstrates that what appears to be a homogeneously high PSF cohort, is, in fact, made of clusters

with distinguishable boundaries, the features of which might allow the identification of phenotypes.

## Summary and future directions

Are there models that explain the pathophysiology of PSF? Yes. Do any of the models predict the possibilities of PSF phenotypes? Yes. Is there evidence to support the existence of PSF phenotypes? Yes. Are they distinguishable? Yes. Do behavioural correlates of function and impairment map onto self-report domain-specific fatigue to allow for distinction between PSF phenotypes? Potentially. Do structural and functional stroke-induced neuronal changes provide a basis for identification of PSF pathophysiology? Yes. Is there evidence of a causal link between neural pathophysiology and PSF? Yes. Do pathophysiological features combined with evidence from causal (interventional) studies indicate the potential presence of phenotypes? Yes. Are variables of a single type (perceptual/behavioural/neuronal) sufficient for the identification of phenotypes, or is it necessary to combine multiple types? Unknown; however, the only phenotyping exercise thus far used a combination of the three types of variable to demonstrate effective clustering.

While there has been significant progress over the last decade in our understanding of PSF, there is a long way to go and any future investigations into pathophysiology and intervention development must carefully identify the aspects of PSF they wish to understand or reverse. Phenotyping provides a systematic approach to identify homogeneous patient groups for understanding PSF pathophysiology. For such phenotyping to avoid pseudo-phenotypes, attention must be paid to constructs such as primary and secondary PSF (Box 1) where the inclusion of secondary PSF would significantly muddy the waters both for progressing our understanding and development of treatments. The recent guidelines for clinical management of post-stroke fatigue provide a comprehensive screening tool that helps with the differential diagnosis of PSF (English et al., 2023) and enable a distinction between primary and secondary PSF. A second construct that will provide a useful boundary within which to restrict any future phenotyping exercise is the distinction between fatigue and fatigability. While those with high fatigue do exhibit diminished behavioural endurance, physiological markers of endurance are unaltered (Rahamatali et al., 2020) in PSF, indicating that reduced behavioural endurance may not result in, but is a result of, post-stroke fatigue.

In summary, phenotyping will be a critical step in developing effective interventions. Understanding the pathophysiology of PSF phenotypes will be imperative for designing phenotype-specific interventions. The sensory attenuation model of PSF provides robust theoretical grounding for the identification of PSF phenotypes.

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

## Additional information

### Competing interests

None declared.

### Author contributions

Sole author.

### Funding

None.

### Acknowledgements

The author thanks the reviewers for highlighting significant shortcomings which resulted in a much-improved version of the manuscript.

### Keywords

fatigue, models, phenotypes, post-stroke fatigue

## Supporting information

Additional supporting information can be found online in the Supporting Information section at the end of the HTML view of the article. Supporting information files available:

**Peer Review History**

