## [Peer Review History · The Journal of Physiology]

Post-stroke fatigue - a multi-dimensional problem or a cluster of disorders? A case for phenotyping post-stroke fatigue

Annapoorna Kuppaswamy
DOI: 10.1113/JP285900

Corresponding author(s): Annapoorna Kuppaswamy (A.Kuppaswamy@leeds.ac.uk)

The following individual(s) involved in review of this submission have agreed to reveal their identity: Sheila Schindler-Ivens (Referee #1); John Harvey Kindred (Referee #2); Jill Stewart (Referee #3)

Review Timeline:	Submission Date:	26-Feb-2024
	Editorial Decision:	25-Apr-2024
	Revision Received:	16-Jul-2024
	Editorial Decision:	30-Jul-2024
	Revision Received:	16-Sep-2024
	Accepted:	10-Oct-2024

Senior Editor: Laura Bennet

Reviewing Editor: Richard Carson

Transaction Report:

Dear Dr Kuppuswamy,

Re: JP-TR-2024-285900 "Post-stroke fatigue - a multi-dimensional problem or a cluster of disorders? A case for phenotyping post-stroke fatigue" by Annapoorna Kuppuswamy

Thank you for submitting your manuscript to The Journal of Physiology. It has been assessed by a Reviewing Editor and by 3 expert referees and we are pleased to tell you that it is potentially acceptable for publication following satisfactory major revision.

Please address all the points raised and incorporate all requested revisions or explain in your Response to Referees why a change has not been made. We hope you will find the comments helpful and that you will be able to return your revised manuscript within 2 months. If you require longer than this, please contact journal staff: jp@physoc.org. Please note that this letter does not constitute a guarantee for acceptance of your revised manuscript.

REVISION CHECKLIST:

We look forward to receiving your revised submission.

Yours sincerely,

Laura Bennet
Senior Editor
The Journal of Physiology

REQUIRED ITEMS

- Please include an Abstract Figure file, as well as the Figure Legend text within the main article file. The Abstract Figure is a piece of artwork designed to give readers an immediate understanding of the Review Article and should summarise the main conclusions. If possible, the image should be easily 'readable' from left to right or top to bottom. It should show the physiological relevance of the Review so readers can assess the importance and content of the article. Abstract Figures should not merely recapitulate other figures in the Review. Please try to keep the diagram as simple as possible and without superfluous information that may distract from the main conclusion of the Review. Abstract Figures must be provided by authors no later than the revised manuscript stage and should be uploaded as a separate file during online submission labelled as File Type 'Abstract Figure'. Please ensure that you include the figure legend in the main article file. All Abstract Figures will be sent to a professional illustrator for redrawing and you may be asked to approve the redrawn figure before your paper is accepted.

- Your MS must include a complete "Additional information section" with the following 4 headings and content:

Competing Interests: A statement regarding competing interests. If there are no competing interests, a statement to this effect must be included. All authors should disclose any conflict of interest in accordance with journal policy.

Author contributions: Each author should take responsibility for a particular section of the study and have contributed to writing the paper. Acquisition of funding, administrative support or the collection of data alone does not justify authorship; these contributions to the study should be listed in the Acknowledgements. Additional information such as 'X and Y have contributed equally to this work' may be added as a footnote on the title page.

It must be stated that all authors approved the final version of the manuscript and that all persons designated as authors qualify for authorship, and all those who qualify for authorship are listed.

Funding: Authors must indicate all sources of funding, including grant numbers. If authors have not received funding, this must be stated.

It is the responsibility of authors funded by RCUK to adhere to their policy regarding funding sources and underlying research material. The policy requires funding information to be included within the acknowledgement section of a paper. Guidance on how to acknowledge funding information is provided by the Research Information Network. The policy also requires all research papers, if applicable, to include a statement on how any underlying research materials, such as data, samples or models, can be accessed. However, the policy does not require that the data must be made open. If there are considered to be good or compelling reasons to protect access to the data, for example commercial confidentiality or legitimate sensitivities around data derived from potentially identifiable human participants, these should be included in the statement.

Acknowledgements: Acknowledgements should be the minimum consistent with courtesy. The wording of acknowledgements of scientific assistance or advice must have been seen and approved by the persons concerned. This section should not include details of funding.

- The reference list must be in alphabetical order, rather than numbered, to comply with our Journal format.

- Author profile(s) must be uploaded via the submission form. Authors should submit a short biography (no more than 100 words for one author or 150 words in total for two authors) and a portrait photograph of the two leading authors on the paper. These should be uploaded and clearly labelled together in a Word document with the revised version of the manuscript. Any standard image format for the photograph is acceptable, but the resolution should be at least 300 DPI and preferably more. A group photograph of all authors is also acceptable, providing the biography for the whole group does not

exceed 150 words.

- Please include a full title page as part of your main article (Word) file, which should contain the following: title, authors, affiliations, corresponding author name and contact details, keywords, and running title.

EDITOR COMMENTS

Reviewing Editor:

Three reviews were received from experts in the field. As you will note, the referees differ somewhat in how they view the clarity of the presentation. Referee #1, in particular, expresses the view that certain key points are not shown to be supported sufficiently by dint of argument or empirical evidence. As these concerns may be representative of some of the readership of the Journal, I would urge that, in revising the manuscript, they are addressed directly. While Referee #2 and Referee #3 were overall more positive in their assessments, both made several important suggestions which, if incorporated, are likely to enhance the impact of this work.

Please also see 'Required Items' above.

Senior Editor:

Thank you for your submission, as noted by the reviewing editor, there is a diversity of opinions by reviewers, and they all offer guidance to be reviewed by the authors. Please take into consideration all of the reviewers comments in your revision to ensure robustness of scientific interpretations and perspectives

REFeree COMMENTS

Referee #1:

This is a position paper written to build the case that fatigue in people who have had a stroke is not a single condition and that the phenomenon that we call fatigue is likely several different phenomena. There is a secondary argument that different manifestations of fatigue may be the result of a single primary deficit, namely a failure to adequately attenuate sensory input. While an interesting topic, this particular manuscript lacks adequate rigor to advance the conversation around this issue. Weaknesses include but are not limited to 1) no definition of fatigue, cluster, or phenotype; 2) no description or characterization of the phenomenon of interest (i.e. post-stroke fatigue); 3) no clearly articulated rationale (other than the thalamus is often involved) for aberrant gain control as a leading explanation for post-stroke fatigue. There is a smattering of literature presented that does not squarely support the author's argument and does not defend the list of conclusions in the summary (presented in Q & A format) at the end of the article. No alternative hypotheses or detracting arguments are presented.

Impact on the area of research

Low

Insight into physiological mechanisms in this field

Low. Rigorous arguments in support of "phenotypes" and their underlying mechanisms were not made.

Originality of the research

N/A, This is a position paper, not a research paper.

Study design and robustness of the experimental data

N/A, This is a position paper.

Validity of the conclusions

Low. Conclusions are not supported strongly by evidence from the literature.

Referee #2:

The author has performed a topical review on post-stroke fatigue (PSF). The author is one of the leading experts in this field, and her knowledge of the subject is apparent and appropriate for this invited review. In this review, the author also questions if and/or how investigators and clinicians may better classify PSF based on contributing factors and possible mechanisms. It seems most of the phenotyping is in relation to the sensory-attenuation model of PSF. Not many experimental studies use this model, although I believe the model/theory is sound. I feel it may be appropriate to incorporate this model into the article's title as about 1/2 the manuscript refers to it in some way. Another point that the author may want to distinguish is between fatigue measured in the sub-acute/acute phases versus the chronic phase. I believe the author is discussing PSF more as it relates to the chronic phase. Readers naïve to fatigue in neurological conditions may not fully understand the major physiological changes in each phase and how those changes may relate to fatigue measured at different time points. I have a few minor comments that the author may want to consider below,

Minor Comments

Line 18 - I would add Skau et al. 2021 (<https://doi.org/10.3389/fpsyg.2021.739764>) and discuss in a bit more detail the difference between fatigue and fatigability. Enoka from the Ref 4 also discusses this in PMID: 34583577 as related to MS.

Line 50-59 - My perception is the way this section is written; possible phenotypes are suggested within the sensory-attenuation model and not between other possible causes of PSF. Would this be a good section to introduce other possible PSF contributors and suggest phenotypes based on other models, including the SAM? (inflammation, GABAergic, glutamatergic, dopaminergic, depressive vs. anxiety phenotype)

Line 82 - does early fatigue mean during the sub-acute and acute phases, or something else? It may be worth setting this review up to discuss PSF in chronic stroke.

Referee #3:

This paper outlines a perspective on a complex health related issue, post-stroke fatigue. The paper provides a nice review of the current status of research on the mechanisms of post-stroke fatigue and offers a perspective on the use of both self-report measures and measures of brain structure and function to help define fatigue in this population. Creation of

subgroups based on phenotypes is an interesting and needed idea that will help address this complex issue.

The Introduction sets up the idea of fatigue phenotypes based on changes in sensory attenuation gain that is expanded upon later in the manuscript. Two primary comments:

The section on neural structure and function integrates with the main idea of changes in sensory gain. However, the section on perception and behavior does not clearly correspond to this perspective. This section focuses mostly on cognitive versus physical aspects of fatigue and the relationship between fatigue and mood (anxiety, depression). Clearer linkage between this section and the proposed phenotypes later in the manuscript would strengthen the paper.

In the section on the sensory attenuation model, three potential phenotypes are presented (interoceptive, exteroceptive, proprioceptive). Text in the next paragraph discusses possible related neural mechanisms for these phenotypes. As a reader, I am following the discussion of the thalamus in this context, however, the discussion of the sensorimotor cortex is a bit more difficult to follow (section starting "Phenotype specific altered gain control mechanisms are likely centred on sensorimotor cortices which can be explored using non-invasive brain stimulation methodologies"). I understood this to be in relation to the three previously defined potential phenotypes, but it is not clear how the sensorimotor cortex would relate to all three phenotypes (for example, the interoceptive phenotype). Later in the same paragraph, there is reference to a 'sensorimotor' phenotype but it is not clear how this relates to three phenotypes described earlier. Clarification in this section would strengthen the author's argument.

Specific Comments:

Perception and behaviour-based clustering of post-stroke fatigue, 1st paragraph

"Self-report is the gold standard measure of any symptom, and fatigue is no exception."

This idea the self-report is the 'gold standard' likely depends on what one means by the term 'symptom'. Some might consider other measures as the gold standard for 'weakness', for example. Suggest a clearer definition for clarity.

Perception and behaviour-based clustering of post-stroke fatigue, 1st paragraph

"...making self-report evidence unsuitable for identification of poststroke fatigue clusters"

Then go onto say later in the same paragraph

"...there is enough evidence of variability in self-report to warrant prospectively designed self-report based studies for cluster identification."

This seems to be offering opposite views on the use of self-report measures to create clusters - one stating that self-report measures should not be used while the other stating that self-report measures should be used. Clarification is suggested.

Summary and future directions

"The construct of primary and secondary fatigue is an important distinction that allows for choosing of relevant variables for phenotyping."

It is not clear what is meant by primary and secondary fatigue - does the author mean fatigue attributable to stroke versus other fatigue discussed in the next sentence?

END OF COMMENTS

Response to reviewers

As the changes made to the manuscript are extensive, I have refrained from providing a point-by-point response to the reviewers' comments. Nonetheless, I have incorporated all requests from all three reviewers. I would like to extend my sincere apologies for submitting a significantly underdeveloped manuscript in the first round [not an excuse but an explanation is that I was moving institutions and cities while submitting this paper, and didn't realise how stressed I was, until I reread my original submission!].

Thanks to the first reviewer for being a harsh but fair critic, thanks to the second reviewer for being kind and conveying the same message of inadequateness but with a lot less harshness, and finally, thanks to the 3rd reviewer for picking up on points not picked up by the other 2 reviewers. Thanks to the editor for choosing a good range of reviewers.

The manuscript is almost completely rewritten (<10% overlap with previous version). I have included several new sections including discussing other models of fatigue, how they compare for purposes of identifying phenotypes and what the phenotypes are (previous word count 2800, now it is 5000). I have also expanded the literature cited and updated it for until June 2024 (previous version had 51 citations, now 93). But crucially, every single section starting from the introduction now has, I believe, an improved clarity in expression of ideas, and includes well-substantiated arguments. While there are still several unresolved issues, I don't think it is due to poor writing, but am open to any further improvements if you feel necessary. I have also now included a Box with essential definitions and a Figure for how phenotypes emerge from the sensory attenuation model.

Dear Dr Kuppuswamy,

Re: JP-TR-2024-285900R1 "Post-stroke fatigue - a multi-dimensional problem or a cluster of disorders? A case for phenotyping post-stroke fatigue" by Annapoorna Kuppuswamy

Thank you for submitting your manuscript to The Journal of Physiology. It has been assessed by a Reviewing Editor and by 3 expert referees and we are pleased to tell you that it is potentially acceptable for publication following satisfactory major revision.

Please address all the points raised and incorporate all requested revisions or explain in your Response to Referees why a change has not been made. We hope you will find the comments helpful and that you will be able to return your revised manuscript within 2 months. If you require longer than this, please contact journal staff: jp@physoc.org. Please note that this letter does not constitute a guarantee for acceptance of your revised manuscript.

REVISION CHECKLIST:

We look forward to receiving your revised submission.

Yours sincerely,

Laura Bennet
Senior Editor
The Journal of Physiology

EDITOR COMMENTS

Reviewing Editor:

The significant improvements that have been realised through this major revision - a substantial rewrite, are acknowledged by all three referees. Nonetheless, all highlight ways in which the manuscript could be improved further. Given the potential impact of the work - which is also emphasised by all referees, I believe that further refinements will ultimately bear dividends.

As Referee #1 provided comments in Confidential Comments for the Editor, I provide the gist of these.

1) It would be valuable to provide clear and crisp definitions of all key terms, such that the scope of each term and the conceptual distinction between terms can be readily understood. For example, can an operational definition of fatigue be provided in conjunction with Box1? It would also seem important to explain which among the terms physical fatigue, mental fatigue, fatigability, fatigue as a perceptual experience, and cognitive, motor, and motivational triggers can be considered equivalent or overlapping and which are conceptually distinct.

2) The organisational structure (both local and global) of the piece might be enhanced, At a global level, it is a concern that the initial emphasis is upon matters such as fatigue vs. fatigability, whereas the motivating topic is post-stroke fatigue. At a local level, there would appear to be scope to craft the presentation such that "run-on sentences" are avoided, along and with "parenthetical insertions". In this context, perhaps some long paragraphs could be split, with the point being conveyed by the resulting elements being thereby highlighted more effectively.

In summary, there is consensus that the topic tackled in this ambitious review is important and likely to be of great interest to many readers. Given its importance and the promise that this submission so clearly offers, some further crafting of the presentation seems particularly worthwhile.

Senior Editor:

Thank you for the substantial changes made in your first revision. This has been well appreciated by the reviewers, who continue to note this is an important area for review and that the review should attract many readers. The reviewers offer further suggestions that will help further refine the review .

REFEREE COMMENTS

Referee #1:

'Please see comments to the editor.'

NOTE FROM EDITORIAL OFFICE: These have been summarised by the Reviewing Editor above.

Referee #2:

The author has returned, for all intents and purposes, a brand new manuscript. I would like to thank the author for acknowledging the previous criticisms and shortfalls of the previous version. This new attempt is a much better review on the topic of PSF. The author highlights greater breadth of previous research and how that research points to different possible phenotypes of PSF. The expanded length is appropriate for the more elaborate and detailed discussion. I have no major concerns with the manuscript in its current state.

My only minor criticism is that the story of neuroinflammation in the chronic stage of stroke is largely incomplete. While the referenced study by Jolly et al. 2024 is informative it does not fully explore the relationship between peripheral and central inflammation. As stroke survivors age it is not out of the question that neurodegeneration and peripheral inflammation could increase, especially when considering comorbid factors which may have been present at the time of stroke, e.g., obesity. I do have some bias in this area as this is an area I am actively investigating but I believe central/neuro-inflammation may play a bigger role in PSF than the author alludes to. However, there is a large gap of knowledge in this area and the possible connection in fatigue in chronic stroke should also be tempered until more evidence supports this.

Referee #3:

I appreciate the author's extensive revision of this manuscript. The overall flow and focus are much improved and it is clear that an argument for phenotyping of PSF is being made.

The final sentence of the manuscript states "The sensory attenuation model of PSF provides robust theoretical grounding for identification of PSF phenotypes." The author should consider more explicitly making this argument throughout the manuscript. As currently written, emphasis on the potential benefits of this model for the development of phenotypes does not clearly come through. This would mostly require relatively small edits with clearer, more explicit language.

Model suitability for phenotype classification

Overall, it appears the author is arguing that the sensory attenuation model is best suited for defining phenotypes in chronic fatigue - but the author could state this more explicitly. Additionally, a reorganization of the paragraph is suggested. The flow and organization might be improved if the arguments against certain models for phenotyping occur first, followed by a clearer discussion for the sensory attenuation model (versus the switching back and forth). Reference to the figure of the model may also be appropriate here.

In the section 'Proposed PSF phenotypes', three potential phenotypes are described with an added figure. The next section 'Sensory attenuation, thalamocortical circuits and PSF' references these phenotypes. However, the following three sections discuss other phenotypes and do not clearly refer to these 3 phenotypes or the sensory attenuation model. This may have been purposeful, however, more clearly stating that these are alternatives would be helpful. Alternatively, providing a link back to the outlined phenotypes/model would be helpful.

END OF COMMENTS

1st Confidential Review

16-Jul-2024

Reviewing Editor:

The significant improvements that have been realised through this major revision - a substantial rewrite, are acknowledged by all three referees. Nonetheless, all highlight ways in which the manuscript could be improved further. Given the potential impact of the work - which is also emphasised by all referees, I believe that further refinements will ultimately bear dividends.

Response: Thanks again to all three referees and editors for such an encouraging response to the re-write. I have now further crafted the essay to make it eminently more readable and hopefully more informative than the previous version. I have tracked changes throughout the manuscript and am refraining from copy pasting all changes in this document, but summarising the changes made.

As Referee #1 provided comments in Confidential Comments for the Editor, I provide the gist of these.

1) It would be valuable to provide clear and crisp definitions of all key terms, such that the scope of each term and the conceptual distinction between terms can be readily understood. For example, can an operational definition of fatigue be provided in conjunction with Box1? It would also seem important to explain which among the terms physical fatigue, mental fatigue, fatigability, fatigue as a perceptual experience, and cognitive, motor, and motivational triggers can be considered equivalent or overlapping and which are conceptually distinct.

Response: I have now included more definitions within the box to include all debatable terminology used in the manuscript. This is primarily concerned with the 'dimensions of fatigue' section. I have also, in conjunction, made alterations to the main script to make this section less ambiguous.

2) The organisational structure (both local and global) of the piece might be enhanced, At a global level, it is a concern that the initial emphasis is upon matters such as fatigue vs. fatigability, whereas the motivating topic is post-stroke fatigue. At a local level, there would appear to be scope to craft the presentation such that "run-on sentences" are avoided, along and with "parenthetical insertions". In this context, perhaps some long paragraphs could be split, with the point being conveyed by the resulting elements being thereby highlighted more effectively.

Response: I have now introduced stroke and fatigue right from the beginning of the manuscript to build on the stroke story steadily rather than to drop it in midway. I have introduced more paragraphs at the appropriate junctures. I have tried my best to split run-on and parenthetical sentences. However, this, and other changes has now resulted in a manuscript that is over 6000 words long!

In summary, there is consensus that the topic tackled in this ambitious review is important and likely to be of great interest to many readers. Given its importance and the promise that this submission so clearly offers, some further crafting of the presentation seems particularly worthwhile.

Senior Editor:

Thank you for the substantial changes made in your first revision. This has been well appreciated by the reviewers, who continue to note this is an important area for review and that the review should attract many readers. The reviewers offer further suggestions that will help further refine the

review.

Referee #2:

The author has returned, for all intents and purposes, a brand new manuscript. I would like to thank the author for acknowledging the previous criticisms and shortfalls of the previous version. This new attempt is a much better review on the topic of PSF. The author highlights greater breadth of previous research and how that research points to different possible phenotypes of PSF. The expanded length is appropriate for the more elaborate and detailed discussion. I have no major concerns with the manuscript in its current state.

My only minor criticism is that the story of neuroinflammation in the chronic stage of stroke is largely incomplete. While the referenced study by Jolly et al. 2024 is informative it does not fully explore the relationship between peripheral and central inflammation. As stroke survivors age it is not out of the question that neurodegeneration and peripheral inflammation could increase, especially when considering comorbid factors which may have been present at the time of stroke, e.g., obesity. I do have some bias in this area as this is an area I am actively investigating but I believe central/neuro-inflammation may play a bigger role in PSF than the author alludes to. However, there is a large gap of knowledge in this area and the possible connection in fatigue in chronic stroke should also be tempered until more evidence supports this.

Response: This is a valid observation. In the interests of not lengthening the manuscript even more, and shifting focus away from neural mechanisms and phenotypes, but at the same time acknowledging the huge gap in this area, I have now included a sentence where Jolly 2024 is cited, to suggest there is still vast scope to examine links with ongoing inflammation and fatigue. I hope this is sufficient.

Referee #3:

I appreciate the author's extensive revision of this manuscript. The overall flow and focus are much improved and it is clear that an argument for phenotyping of PSF is being made.

The final sentence of the manuscript states "The sensory attenuation model of PSF provides robust theoretical grounding for identification of PSF phenotypes." The author should consider more explicitly making this argument throughout the manuscript. As currently written, emphasis on the potential benefits of this model for the development of phenotypes does not clearly come through. This would mostly require relatively small edits with clearer, more explicit language.

Response: I am grateful to you for having highlighted this shortcoming. This style of writing comes from being trained by 'old school' British physiologists who have always emphasised the need to be understated. Combine this advice with a field that is mired with uncertainties, one ends up not conveying what one means! Given the two options of either altering the conclusions to be less strong, or making edits to the language used, I have now chosen to retain the conclusion as I do think there is strong enough evidence to support the conclusion and have now made changes throughout and inserted more definitive summary statements in several sections where I felt they were appropriate.

Model suitability for phenotype classification

Overall, it appears the author is arguing that the sensory attenuation model is best suited for defining phenotypes in chronic fatigue - but the author could state this more explicitly. Additionally, a reorganization of the paragraph is suggested. The flow and organization might be improved if the arguments against certain models for phenotyping occur first, followed by a clearer discussion for the sensory attenuation model (versus the switching back and forth). Reference to the figure of the model may also be appropriate here.

Response: Thank you for this suggestion. I have now restructured this section as suggested and it reads a lot better with a much stronger argument in favour of the sensory attenuation model.

In the section 'Proposed PSF phenotypes', three potential phenotypes are described with an added figure. The next section 'Sensory attenuation, thalamocortical circuits and PSF' references these phenotypes. However, the following three sections discuss other phenotypes and do not clearly refer to these 3 phenotypes or the sensory attenuation model. This may have been purposeful, however, more clearly stating that these are alternatives would be helpful. Alternatively, providing a link back to the outlined phenotypes/model would be helpful.

Response: You are correct in thinking the lack of reference to phenotypes is deliberate. However, on rereading it, it is possible to make a case for linking it back to the proposed phenotypes which I have now done. However, it is also appropriate to entertain the possibility of other phenotypes, which I have now done, but also mentioned why I think this is less likely in the thalamocortical circuits section.

Dear Dr Kuppuswamy,

Re: JP-TR-2024-285900R2 "Post-stroke fatigue - a multi-dimensional problem or a cluster of disorders? A case for phenotyping post-stroke fatigue" by Annapoorna Kuppuswamy

We are pleased to tell you that your paper has been accepted for publication in The Journal of Physiology.

Authors should note that it is too late at this point to offer corrections prior to proofing. Major corrections at proof stage, such as changes to figures, will be referred to the Editors for approval before they can be incorporated. Only minor changes, such as to style and consistency, should be made at proof stage. Changes that need to be made after proof stage will usually require a formal correction notice.

Yours sincerely,

Laura Bennet
Senior Editor
The Journal of Physiology

P.S. - You can help your research get the attention it deserves! Check out Wiley's free Promotion Guide for best-practice recommendations for promoting your work at www.wileyauthors.com/eeo/guide. You can learn more about Wiley Editing Services which offers professional video, design, and writing services to create shareable video abstracts, infographics, conference posters, lay summaries, and research news stories for your research at www.wileyauthors.com/eeo/promotion.

IMPORTANT NOTICE ABOUT OPEN ACCESS: To assist authors whose funding agencies mandate public access to published research findings sooner than 12 months after publication, The Journal of Physiology allows authors to pay an Open Access (OA) fee to have their papers made freely available immediately on publication.

You can check if your funder or institution has a Wiley Open Access Account here: <https://authorservices.wiley.com/author-resources/Journal-Authors/licensing-and-open-access/open-access/author-compliance-tool.html>.

EDITOR COMMENTS

Reviewing Editor:

Two of the three original referees have assessed this second revision of the submission. Both are of the opinion that the manuscript has been improved further. I share this view, and concur with the referees that this review has the potential to make a valuable contribution to the literature.

REFeree COMMENTS

Referee #2:

I have no major or minor concerns with the manuscript in its current form.

Referee #3:

The author has further focused and streamlined the manuscript in this version. Overall, key points are clearer and the flow is stronger. This is a nice paper on an important topic. I have no further comments.

Note that the author should carefully edit the final version for grammar and sentence structure prior to publication.

2nd Confidential Review

16-Sep-2024